# Sentinel-1 Interferometry and UAV Aerial Survey for Mapping Coseismic Ruptures: Mts. Sibillini vs. Mt. Etna Volcano

**Marco Menichetti** [1,2,*] **, Matteo Roccheggiani** [1] **, Giorgio De Guidi** [2,3] **, Francesco Carnemolla** [3] **, Fabio Brighenti** [3] **, Giovanni Barreca** [2,3] **and Carmelo Monaco** [2,3,4]

1. Dipartimento di Scienze Pure ed Applicate, Università di Urbino, 61029 Urbino, Italy
2. CRUST—Interuniversity Center for 3D Seismotectonics with Territorial Applications, 66100 Chieti Scalo, Italy
3. Dipartimento di Scienze Biologiche Geologiche e Ambientali, Università di Catania, 95125 Catania, Italy
4. Istituto Nazionale di Geofisica e Vulcanologia, Osservatorio Etneo—Sezione di Catania, 95125 Catania, Italy
* Correspondence: marco.menichetti@uniurb.it

**Abstract:** The survey and structural analysis of surface coseismic ruptures are essential tools for characterizing seismogenic structures. In this work, a procedure to survey coseismic ruptures using satellite interferometric synthetic aperture radar (InSAR) data, directing the survey using Unmanned Aerial Vehicles (UAV), is proposed together with a field validation of the results. The Sentinel-1 A/B Interferometric Wide (IW) Swath TOPSAR mode offers the possibility of acquiring images with a short revisit time. This huge amount of open data is extremely useful for geohazards monitoring, such as for earthquakes. Interferograms show the deformation field associated with earthquakes. Phase discontinuities appearing on wrapped interferograms or loss-of-coherence areas could represent small ground displacements associated with the fault's ruptures. Low-altitude flight platforms such as UAV permit the acquisition of high resolution images and generate 3D spatial geolocalized clouds of data with centimeter-level accuracy. The generated topography maps and orthomosaic images are the direct products of this technology, allowing the possibility of analyzing geological structures from many viewpoints. We present two case studies. The first one is relative to the 2016 central Italian earthquakes, astride which the InSAR outcomes highlighted quite accurately the field displacement of extensional faults in the Mt. Vettore–M. Bove area. Here, the geological effect of the earthquake is represented by more than 35 km of ground ruptures with a complex pattern composed by subparallel and overlapping synthetic and antithetic fault splays. The second case is relative to the Mt. Etna earthquake of 26 December 2018, following which several ground ruptures were detected. The analysis of the unwrapped phase and the application of edge detector filtering and other discontinuity enhancers allowed the identification of a complex pattern of ground ruptures. In the Pennisi and Fiandaca areas different generation of ruptures can be distinguished, while previously unknown ruptures pertaining to the Acireale and Ragalna faults can be identify and analyzed.

**Keywords:** coseismic ruptures; InSAR; photogrammetry; UAV

## 1. Introduction

The detection of the spatial geometric attitude and the structural analysis of surface small-scale coseismic ruptures are essential to characterize seismogenic structures [1,2].

The localization of coseismic surface ruptures and the collection of data are usually carried out with a classical field survey that, in steep and morphologically complex mountain regions, is time demanding and highly consuming in term of human and economic resources [3–6].

An important challenge in detecting and measuring coseismic surface ruptures in the field is represented by the easy and rapid erosional degradation of fault scarps, especially in the mountain range where storms are frequents and the snow cover can reach a few meters [7]. The degradation of exposed coseismic rupture surfaces begins immediately

after an earthquake, and within a few weeks many important structural relationships of the surface rupture can be completely erased [7]. The main challenge is to rapidly examine coseismic ruptures that normally extend for tens of kilometers across regional structural domains and consist of complex shear zones [8]. Locating them in the field is a priority for geologists in order to proceed with a detailed high resolution investigation at the highest level of detail to avoid the oversimplification of geological structures [5,9].

The use of satellite interferometric synthetic aperture radar (InSAR) with high sensitivity to displacements facilitates the localization of the coseismic crustal deformation [6,10]. The general deformation fringe patterns in the satellite's ground line of sight (LOS) highlight minor surface ruptures, indicating possible fault reactivations and contributing to a better understanding of the coseismic events [11]. Despite the need of a few days to collect satellite data on the same area, interferometry is a valuable tool to guide a field survey and locate the area in which to collect more precise data from near-field sensors [12–16].

In recent years, geological surveying in the field has benefited from the new digital revolution, reaching a mature procedure that continuously improves in its efficiency [17–19]. The available technology allows for the easy acquisition of spatial data, using different methodological approaches and data sources at variable scales [20]. The availability of a precise location with the Global Navigation Satellite System (GNSS), and the accessibility of high resolution multitemporal satellite images, digital aerial photogrammetry, terrestrial and aerial Lidar techniques, allows the production of precise and high resolution topographic maps which make it is easier to report geological structures and landforms in detail [21].

The main objectives of this paper are concerned with the verification of a procedure that combines two methodologies of analysis with a final field validation of results. The aim is to contribute to the improvement of the knowledge about the seismic hazard of the study areas. The interferometric data permit the quick identification of where the primary coseismic ruptures are located. In the identified regions, it is possible to carry out a survey at a higher resolution using low-altitude flight platforms such as Unmanned Aerial Vehicles (UAV). We tested the method in two distinct areas where earthquakes with coseismic ruptures recently occurred: Mts. Sibillini, in central Italy, where a sequence of strong earthquakes (Mw > 6) with extensional kinematics occurred in August–October 2016 [22], and Mt. Etna volcano in Sicily, where a shallow and low–moderate magnitude earthquake (Mw~5) with strike-slip kinematics, associated with a summit eruption, occurred on 26 December 2018 [https://www.ct.ingv.it/index.php/monitoraggio-e-sorveglianza/banche-dati-terremoti/terremoti] (accessed on 15 March 2023) (Figure 1). The two regions are quite different in terms of geological features and seismotectonic; moreover, central Italy is a mountain area, sparsely populated, while M. Etna's slopes have a much higher population density and man-made structures.

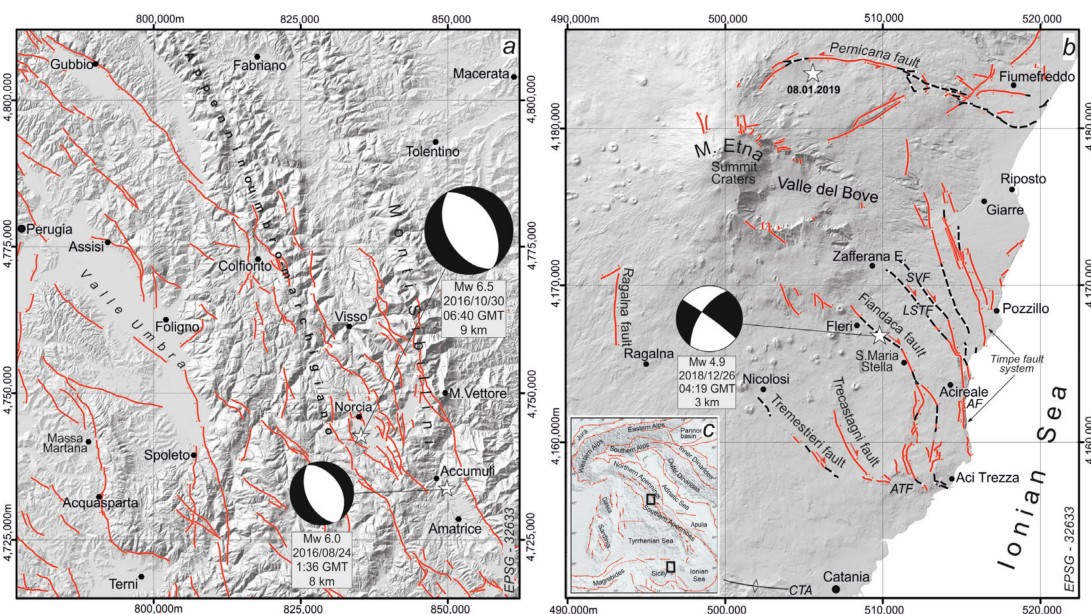

**Figure 1.** Structural framework of the investigated areas. (**a**) Southern portion of the Umbria-Marche Apennines, with Quaternary extensional fault systems and focal mechanism of the largest earthquakes which occurred between August and October 2016 (stars indicate the epicenters)—modified from [23]; (**b**) south-eastern flank of Mt. Etna volcano, with the main fault systems and the focal mechanism of the large seismic event which occurred on 26 December 2018; Acireale fault (AF), Linera-Santa Tecla fault (LSTF), Santa Venerina fault (SVF), Aci Trezza fault (ATF) and Catania Terreforti anticlines (CTA) (star indicate the epicenter)—modified from [24]; (**c**) inset shows the geodynamic context of the Italian Peninsula and locations of the investigated areas.

## 2. Materials and Methods

### 2.1. Workflow

Interferometric images can be used as a guide to field work focused to the detection of small-scale fault ruptures related to moderate–large earthquakes [6,25,26]. At larger topographic scales, the detailed survey of fault geometries, displacements and geological offsets needs near field sensors [27]. Flying platform systems which are ready-to-use, such as small UAVs able to carry different types of payload, including sensors and cameras, allow the acquisition of georeferenced optical (RGB) low altitude imagery, both zenithal and oblique [28,29]. These images can be processed with photogrammetric computer vision algorithms to build true precision 3D models with centimeter accuracy. Highly detailed topographic maps are the direct products, but the possibility of analyzing geological structures in hazardous or inaccessible areas represents one of the most important applications [30,31].

The proposed methods (Figure 2) for locating and mapping coseismic ruptures at the finest details combine different procedures of analysis of satellite and low-altitude images. The workflow consists of a combination of interactive steps, where structural analysis can be performed over different virtual objects from many viewpoints, repeatedly and practically at low cost. This allows the rapid evaluation of the geometry of the structures with the possibility of further analysis and interpretation by geologists with different expertise. The procedure starts with the processing of interferometric data (Phase 1—Figure 2) to infer ground ruptures and report them on a map. The obtained map and the related satellite interferometry products can be compared with geological and topographic maps available for the region. This allows the targeting of the area where higher resolution mapping is required, using low-altitude flights from UAVs (Phase 2—Figure 2). The acquired zenithal and oblique images can be processed with Structure from Motion (SfM) algorithms to obtain point clouds, 3D models, orthomosaic images and digital surface models (DSM), and to perform spatial and geometrical analyses (Phase 3—Figure 2). In

the field, the observable geometrical parameters of the surface rupture zones are localized along the strike using GNSS devices. The geological field survey was performed through Fieldmove software v1.5.2.152001 (Petroleum Expert Edinburgh, Scotland, UK—https: //www.petex.com/products/move-suite/digital-field-mapping/ (accessed on 15 March 2023) installed on an Apple IPad-Pro. The observable parameters that define rupture are surveyed, including the lithology, the thickness, the geometric arrangement of different fracture sets, the patterns of splaying and the degree of interconnectivity. On the scarp surface, the offset (Sd), the vertical (Vd) and horizontal displacement (Hd) are recorded, including, where possible, the slip vectors. Finally, image processing products and field survey checks allow the drawing of a detailed and accurate map of coseismic ruptures, and for the geometrical and structural kinematic analyses to be carried out (Phase 4–Figure 2).

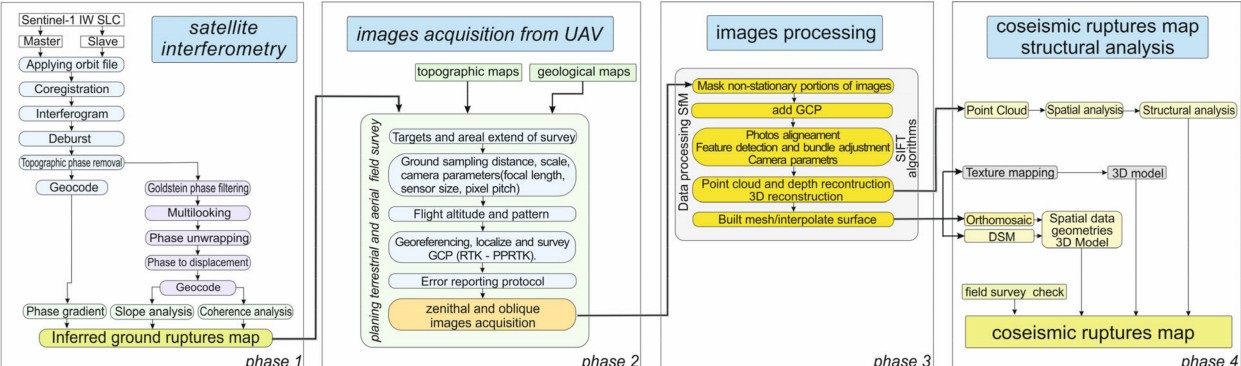

**Figure 2.** Workflow design consisting of the following phases: (**1**) satellite interferometry analysis for the rupture recognition; (**2**) acquisition of large-scale images from UAV; (**3**) checking of images' quality and image data processing; and (**4**) operations on processed layers and landforms (3D objects), and data extraction to produce the map.

## *2.2. Satellite Interferometry*

The Sentinel-1 mission of the European Space Agency (ESA) offered the possibility of acquiring SAR images with a minimum revisit time of six days over Europe with the interferometric Wide (IW) Swath TOPSAR (Terrain Observation with Progressive Scans SAR) mode [32], until 2021. This huge volume of open data is useful for monitoring hazardous events, such as earthquakes, with the production and analysis of differential interferograms (D-InSAR) [33].

Satellite interferometry was first used in 1992 to map the coseismic displacement of the Landers earthquake (M 7.3) [11]. Using a pair of SAR images, one acquired before the seismic event and the other immediately after, interferometry can be used to map the fault geometry, to define the dynamics of ruptures and the seismic strain field, and to estimate the epicentral location [34]. Interferograms of a moderate–large earthquake show the deformation field caused by the movement of the principal seismogenic fault, but also of minor related structures. Price and Sandwell [35] used surface displacement gradient maps of complex interferograms to visualize lineaments. Guerrieri et al. (2010) [36] and Fukushima et al. (2013) [37,38] have shown that phase discontinuities appearing on wrapped interferograms could represent small ground displacements. Fujiwara et al. (2016) [37] mapped linear surface ruptures of the 2016 Kunamoto seismic sequence using ALOS-2 SAR interferometry, which is advantageous for detecting ground displacements due the high coherence achievable by L-band.

Differential interferometric analysis (D-InSAR) has been conducted using open source SNAP software (Sentinel Application Platform) [39], which provides a useful suite of indispensable tools for this purpose. The radar images to be processed have been downloaded from the official ESA website, called Sentinel Open Hub [https://scihub.copernicus.eu/] (accessed on 15 March 2023). In total, 6 pairs of images (4 for central Italy, 2 for Sicily) were processed in Interferometric Wide (IW) Swath TOPSAR format. Each pair covers the

pre- and post-seismic period, related to each main shock. The obtained interferograms were analyzed with different techniques to identify possible coseismic fractures. The whole classic interferometric process is summarized in Figure 2.

Under some assumptions [36–38], acquisitions with different geometry can be combined to measure the vertical and horizontal components of motion. The produced interferograms, unwrapped and with phase converted to displacement, were imported into a GIS environment. Through a raster calculation tool, knowing the respective angles of incidence and azimuth LOS, it was possible to quickly combine the respective ascending and descending acquisitions. In this case, to extract the coseismic breaks, the study focused on the vertical component of the movement (up–down) since this seems to better highlight the discontinuities, preserving the data of both acquisitions. The final step for identifying the coseismic breaks was taken by calculating a slope map, filtering the result for the inclination values greater than 45°. This filter operation was necessary to eliminate, as much as possible, noise, errors to be discarded and problems deriving from the reference DTM.

The subsequent analysis, focused on identifying and mapping the coseismic ruptures, was conducted in a GIS environment. Many discontinuities were easily detectable directly from the wrapped phase thanks to the high final coherence, especially in the case of Mt. Etna. However, several approaches have been tried to improve the detection of minor ruptures; the linear areas characterized by a loss of coherence were referred to possible coseismic ruptures and identified as faults. Furthermore, the displacement field was analyzed using classical morphological analysis methods such as the slope map. This approach has been found to be very efficient, even though it remains susceptible to unwrapping errors. For this reason, the phase gradient of each interferogram was also corrected directly using the real and imaginary values of the interferometric phase, following the method suggested by Price and Sandwell 1998 [35].

### 2.3. UAV Acquisition and Image Processing

In recent years, ready-to-use flying platform systems such as small UAVs have been an important tool for acquiring high-resolution images at low costs [28]. The acquired images, both zenithal and oblique, processed with a new generation of image analysis algorithms such as "Structure from Motion" (SfM), developed in computer vision software, allow 3D spatial geolocalized clouds of data with centimeters of accuracy to be obtained. Topography or DSM maps and orthomosaic images are the direct products of this technology, with the possibility of analyzing the spatial geological structures from many viewpoints. During the UAV survey, each image recorded the georeferenced data from the onboard GNSS sensor. Generally, these coordinates had an accuracy of few meters (2–5 m), and therefore precise georeferenced and well-distributed GCP (Ground Control Point) needs to be used to optimize the bundle adjustment, reaching a sub-centimeter accuracy [40].

Considering the inability of the RGB camera to penetrate the vegetation (grass and trees), the terrain model could include the surface (DSM) but with potential errors in the dataset. For this reason, improvements in photogrammetric algorithms have subsequently enhanced the accuracy of DSMs created from aerial imagery in geological applications [29].

The measurement of coseismic ruptures ranging in size from cm to dm requires large-scale images with suitable pixel resolution. The ground sample distance (GSD) is the linear dimension of the footprint of a pixel of the sample image on the ground, assuming near-vertical imagery [30]. In image acquisition planning, the determination of the size of geological features of interest is an important early step for the photogrammetric survey design. In the following, the key parameters are described: (a) the minimum level of detection (MLD), typically on the order of cm to mm, is the ratio between the analyzed feature size and the topographic surface noise (e.g., grass); (b) the pixel resolution, typically dm to cm per pixel side length, is used for characterizing the geological features and is calculated from ground sample distance (GSD). This parameter is directly linked to the distance of the camera from the object and the size of the camera sensor, with respect to the width of digital images in the pixel and the focal length of the camera [4]. Considering

the fixed camera physical parameters (width of sensor and focal length), adjusting the distance-to-subject for the camera is the simplest way to control the GSD. Therefore, the flight height represents a sensible parameter during the survey, especially in a mountainous area with complex topography.

The survey of many km-long strips in steep mountain slopes, where coseismic ruptures have previously been localized, presents several problems for the georeferencing of the images. The use of objects with known dimensions has been preferred in many cases, over positioning and georeferencing the GCP. In a few cases, the derived point clouds have good precision (centimeters) and low accuracy (meters) compared to the geospatial reference frames [30]. For each 3D cloud model, the possible survey errors, including reprojection and camera location, are known and consistent with the size of surveyed features [4]. Several checks and comparison are made between the offset data measured in the field and the same data detected on the point cloud model. The results show errors of a few cm, less than 5%, where the model generally overestimates the real data. Finally, the quality of the topographic and geological 3D point clouds can be improved using several techniques, including editing, resampling and different reprocessing methods or filters.

During the second half of 2016 and 2017, we used two commercial quadcopter drones to acquire more than 15,000 2D zenithal and oblique aerial photos along 35 km of coseismic ruptures in the Mts. Sibillini area. About 5000 oblique and zenithal low-altitude aerial photos were acquired in the Mt. Etna region between the last days of December 2018 and March 2019. In the surveys, the platforms used were equipped with a camera stabilized by a gimbal mount producing geotagged photos in RAW format, georeferenced by a dual constellation GNSS receiver (GPS and GLONASS) located on IMU (Inertial Measurement Unit). Two models of available commercial drone were used: in the years 2016/17 a DJI Phantom 3 Pro was used, equipped with a Sony sensors Exmor of size of 1/2.3" with a calibrated FOV 94°–20 mm f/2.8 lens, which allowed a photo resolution of 12.76 Mpixels. In the years 2018/19 the new model of DJI Phantom 4 Pro was used, equipped with a Sony sensors Exmor of size of 1" with a calibrated FOV 84°–24 mm f/2.8 lens, which allowed a photo resolution of 20 Mpixels. Zenithal and oblique images were acquired with at least 85% front overlap and 70% side overlap, meeting the recommended minimum from the photogrammetric rules. All flights took place in optimal lighting conditions, best around mid-day to minimize shadows [4].

The flights and image capture were handled manually using a DJI mobile app with a combination of the cruise parameters. The zenith and oblique images were acquired with flights at optimal altitude above ground level (AGL) to obtain a spatial resolution GSD ranging from about 1.3 cm/px (30 m AGL with a survey area of 70 × 50 m) to about 4.3 cm/pix (100 m AGL with a survey area of 173 × 130 m). The survey procedure started with the location of the area where the coseismic ruptures had been previously identified by the interferometric analysis (Figure 2). Where possible, GCP targets were placed and georeferenced with a GNSS, using Real Time Kinematic (RTK) or post-processing (PPRTK) procedures or laser telemetry [40].

The digital images were processed using SfM algorithms, obtaining 3D clouds of more than $3 \times 10^7$ points for each area. These point clouds allowed us to generate a fully rendered 3D model, making it possible to extract microtopographic features as well as fracture geometries. Moreover, the comparison of the point clouds generated by the photographs taken astride the main seismic events allowed the topographical changes and, possibly, the fault kinematics to be defined. Moreover, digital surface models and orthophoto mosaics allowed the detection of detailed geological features that could be analyzed from different viewpoints. The structural relationships of the surface ruptures during the different events were systematically mapped at scales smaller than 1:500, and the distribution of all fractures with vertical offset > 2 cm was documented.

## 3. Study Areas

### 3.1. Mts. Sibillini in Central Italy

The Umbria-Marche Apennines (Figure 1a) constitute the southernmost sector of the Northern Apennines, part of the peri-Mediterranean system of the Alpine orogen (Figure 1c) and the result of differential movements between the African and European plates. After an early phase of Mesozoic extension, responsible for the tectono-stratigraphic evolution of the Tethys basins, during the Late Oligocene the Africa–Europe convergence caused the subduction of the Adria microplate under the Corsica–Sardinia block. Since the Early Miocene, an arcuate fold-and-thrusts belt developed, migrating to the NE and progressively involving the Adriatic foreland in the deformation, with propagation of the frontal thrust detachment in the foredeep sectors. From the Middle–Late Miocene and up to the Pleistocene, the tectonic regime of the Apennines was characterized by the coeval occurrence of extension in the orogenic hinterland and shortening at the thrust front, as documented by the ages of involved siliciclastic deposits. Extension continued during the Late Pleistocene–Holocene, and it is still active in the study area and along the whole Apennines chain [41–43]. The boundary between extension and contraction is still controversial, but there is general agreement on the geometry of the main active normal fault systems being roughly oriented NW–SE. The surface traces of distinct faults are well mapped along the whole Northern Apennines (Figure 1), with many stepovers between them [42]. Intramountain basins are distributed from the town of Gubbio, through the Colfiorito sector, reaching the area of Mt. Vettore-Norcia and continuing to the southeast, as far as the town of Amatrice (Figure 1a). These basins are bounded by active SW-dipping normal master faults that cut the compressional structures [42].

A significant sequence of normal faulting earthquakes occurred in the Mts. Sibillini area from August to October 2016. This mountain chain constitutes a thick succession of Meso-Cenozoic limestones and marls, deformed by N–S trending folds and thrust over an Upper Miocene siliciclastic sequence (the Laga Formation), a few thousand meters thick, with an offset of a few km [44]. Mt. Vettore is the highest peak, reaching an altitude of 2476 m. Since the late Pliocene, the entire region has experienced extension accommodated by a set of extensional faults, which crosscut compressional structures with an offset of more than 1 km and form the intermountain basins of Pian Grande and Castelluccio di Norcia. Geodetic data indicate, for this area, an active extension of about 3 mm/y that plays a key role in slope morphogenesis [45]. The normal fault system consists of interlinked NNW–SSE trending segments, dipping to the SW with angles > 50°. Situated on the western side of the chain, the faults have a pronounced morphological feature. A continuous limestone scarp, more than 10 km long and marked by a few-meters-thick shear zone with cataclastic breccia, is well documented in the upper slope of the mountain. In the plain area, a few steps with decimetric offset extend for a few kilometers where debris fan and clastic materials have buried other splays that distribute the fault throw. These SW-dipping normal faults were reactivated during the 2016 seismic sequence.

On 24 August 2016, an earthquake with Mw 6.2 struck the towns of Accumoli and Amatrice [22]. Several ground ruptures with decimetric offset along SW-dipping extensional faults occurred over more than 10 km. Two months later, on 26 October, another mainshock with Mw 5.9 occurred 25 km to the north, near the town of Visso. On 30 October, the largest shock of the sequence (Mw 6.5) occurred in the area between the epicenter of the previous earthquakes (Figure 1). These events reactivated many of the existing faults in the area and produced a complex pattern of NNW–SSE trending ruptures with offsets greater than 1 m, extending over a total length of over 35 km in the Tronto Valley, Castelluccio di Norcia plains and Mt. Vettore–Mt. Porche–Mt. Bove sectors (Figure 3) [44]. In particular, the 24 August event produced an offset of about 0.15 m while the 30 October event produced fault scarps that reached a cumulative throw of 0.8 m. Part of the coseismic ruptures due to the Amatrice-Visso-Norcia seismic sequence have been already mapped and described in several papers, ranging from very preliminary surveys [46–49] up to more detailed studies based on drone survey [27].

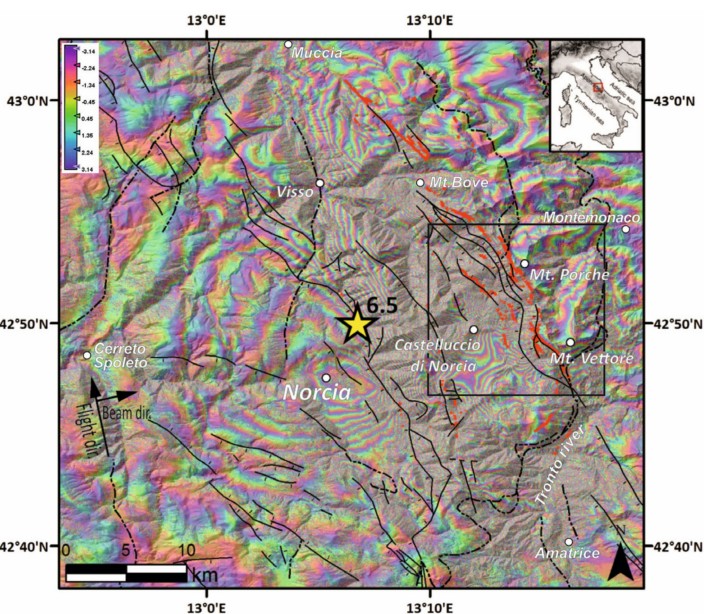

**Figure 3.** Sentinel-1 interferogram (117 ascending orbit) related to the Norcia earthquake (star indicate the epicenter), west of the Mts. Sibillini The main tectonic features from [23] are normal faults (black lines, barbs in the hanging-wall) and thrusts fault (dashed lines). The main coseismic ruptures are indicated in red. Inset frame indicates location of Figure 4.

*3.2. Mt. Etna in Sicily*

Mt. Etna is a composite volcano located in eastern Sicily, along the collisional belt between the African and European plates, where a complex geodynamic framework allows magma to rise from the mantle [50,51]. The volcano currently stands at 3350 m a.s.l., covering an area of 1190 km$^2$ with a baseline circumference of 140 km (Figure 1b). The volcanic activity started about 500 ka ago, mainly with submarine fissural eruptions, but the construction of the current edifice can be traced back to 200 ka ago, with the superposition of several centers [52,53]. The volcanic edifice has grown at the front of the Sicilian-Maghrebian thrust belt and at the northern termination of the Alfeo–Etna Fault System (Figure 1b,c). This is a major kinematic boundary in the western Ionian Sea, associated with the Africa–Europe plate convergence, since it accommodates, by right-lateral kinematics, the differential motion of adjacent western Ionian compartments [54]. Currently, volcanic activity is mainly expressed by weakly explosive activities and abundant lava flows.

The lower eastern flank of Mt. Etna (Figure 1b) is the most tectonically active area of the volcanic edifice both for the number of volcano-tectonic seismic events and for the maximum intensity reached at the epicenter [55]. It is crossed by extensional NNW–SSE to N–S trending faults (the Timpe Fault System) and by NW–SE-oriented dextral strike-slip and oblique structures [56] (Figure 1b), whose intense tectonic activity is evidenced by <6 km deep earthquakes with a magnitude seldom exceeding 4 [57,58]. In general, being very shallow, they cause significant damage relative to the magnitude (epicentral macroseismic intensity up to IX), although in very narrow areas, and are often accompanied by coseismic surface fracturing [58,59]. The Timpe fault system controls the present topography and drainage network of the lower south-eastern flank of Mt. Etna and shows steep escarpments (locally named "Timpe") with very young morphology, mostly Late Pleistocene to Holocene (Figure 1b). The up-to-200-meters-high rectilinear scarp of the Acireale fault controls the Ionian coastline, affecting sedimentary and volcanic rocks ranging in age from Early Pleistocene to historical times [53]. In the south-eastern sector of the volcano, the NW–SE-trending Fiandaca, Linera-Santa Tecla and Santa Venerina faults [60,61] connect by right-lateral motion the Timpe Fault System with the upper slope of the volcano, where the recent eruptive activity has been concentrated along the southern and north-eastern rift zones [50,54,62,63]. The geometry and the sense of movement of most of the main active structures occurring on the eastern flank

of the volcano (Figure 1b) are kinematically consistent, since they accommodate a regional NNW–SSE oriented regional compression and an ~E–W-oriented tension [62,64–67]. In the western flank of the volcano, a prevalent radial deformation, connected to inflation processes, prevails. In the sedimentary basement of the volcano, NNW–SSE-oriented compression has been accommodated by deep thrusting and detachment folding of Pleistocene marine and coastal–alluvial deposits along the southern margin of the volcanic edifice (the Catania and Terreforti anticlines; Figure 1b; [68]).

Structural, morphological and geophysical studies suggest that the eastern flank is also gravitationally spreading towards the sea through the slow motion of several mobile shallow blocks, mostly bounded by tectonic structures (see [69,70] for a review). The sliding sector (Figure 1b) is confined to the west by the NE and S rift zones passing through the summit craters, and to the north and to the south by the left-lateral Pernicana fault and by the right-lateral Aci Trezza fault, respectively [71], which aseismically transfer the extension to the east, towards the offshore [72]. To the west, the N–S striking Ragalna fault, characterized by an oblique-dextral component of motion, has been interpreted as the western boundary of the unstable south flank [73,74]. The rate of the gravitational deformation is one order greater than the tectonic component [60], which, for this reason, could be masked. Nevertheless, uplifted paleo-shorelines at the footwall of normal faults, documented in the SE sector of the volcano [70], suggest that, in the long-term, the tectonic signal prevails over the gravitational signal.

An eruptive phase occurred in the summit area on 24–27 December 2018, with an associated seismic swarm (ca. 250 events with ML < 3; http://sismoweb.ct.ingv.it/index.php (accessed on 15 March 2023)). On 26 December at 02:19 GMT, one of the strongest seismic events ever recorded on Mt. Etna (Mw 4.9; http://cnt.rm.ingv.it/event/21285011 -(accessed on 15 March 2023)) occurred in the south-eastern sector of the volcano, where the Fiandaca fault, along which historical coseismic ruptures have been documented since 1875 [75], was reactivated [24]. This earthquake was triggered by a dyke intrusion in the summit area that caused a significant deformation in the volcanic edifice [76]. The focal mechanism indicated a dextral strike-slip movement with a nodal plane dipping at 88° toward N36° (http://cnt.rm.ingv.it/event/21285011 (accessed on 15 March 2023)). The shallow hypocentral depth (~1 km) of this event caused relevant damage in the urbanized areas of Fleri and Pennisi (Figure 1b). The ground deformation observed by the analysis of interferograms indicated that from 22 to 28 December the entire volcanic edifice was split by the intrusion of a N–S oriented dyke about 6.3 km long, with a depth at its top of 2.5 km and an opening of 1.3 m [77]. The earthquake was accompanied by superficial coseismic ruptures and building damages along the Fiandaca fault, with a deformation zone extending for about 8 km through a complex morphological rural countryside from the Mt. Ilice–Fleri area in the northwest to the Aci Catena–Aci Platani area in the southeast [24,78,79]. After the earthquake, several aftershocks occurred at about 1 km northeast of the northern tip of the Fiandaca Fault, with hypocenters located 1–3 km deep and with focal solutions indicating normal and reverse movements on NE–SW and E–W oriented planes, respectively (http://sismoweb.ct.ingv.it/index.php (accessed on 4 May 2023)). The set of data suggests a kinematic picture consistent with a right simple shear model of deformation, related to the volcano-tectonic framework of the eastern flank of the volcano [24,54].

## 4. Results

### 4.1. Mts. Sibillini Interferogram Analysis

To investigate the surface effects using SAR satellite interferometry, four pairs of Sentinel-1 IW SLC images, related to the two main 2016 seismic events (Amatrice and Norcia), were processed. The interferometric pairs belonged to the descending orbit 22 and ascending orbit 117, acquired immediately before and after the respective events. Figure 3 shows one of the processed interferograms related to the M 6.5 event. Focusing on the area of Mts. Sibillini, the coseismic deformation is distributed in the N–S direction over an area of about 1300 km$^2$, with the major axes of about 55 km from Amatrice to Muccia and

about 45 km between Montemonaco and Cerreto di Spoleto. Considering the extensional structure emerging to the SW of M. Vettore–M. Bove, the deformation extends for about 15 km in the footwall and 30 km in the hanging wall of the major normal fault.

Although characterized by a good basic quality, the interferograms show large areas of decorrelation, both geometric (strong motion) and temporal (vegetation). However, the mountainous morphology also leads to the loss of some information due to shadowing effects. To integrate the information obtained from each interferogram in a single analysis, we followed the strategy of decomposing the components of the movement, combining the results. Differential interferometry provides an estimate of the relative displacement of the earth's surface along the satellite's oblique line of sight (LOS). During the seismic sequences of 24 August and 30 October 2016, the coseismic ground ruptures mostly affected the area of Castelluccio di Norcia and the sector of Mt. Vettore–Mt. Porche–Mt. Bove (Figure 3). Gradient maps allow the highlighting of features with slopes greater than 45 degrees that could be considered coseismic ruptures (Figure 4). Both seismic events showed these gradient lineaments, which were much more extended during the stronger event of 30 October. The ruptures detected in the field survey campaigns during the last months of 2016 and until the fall of 2017 are drawn with a blue line in Figure 4b. The discontinuities extracted from the interferograms can indistinctly represent both the actual superficial faulting of the ground and any newly formed landslide related slopes. The Sentinel-1 results concerning the Amatrice event show quite accurately the deformation that occurred along the western flank of Mt. Vettore. The surface effects of the earthquake were more than 5 km of ground ruptures, aligned along the well-known system of normal faults of Mt. Vettore. The slope map calculated on the vertical motion displacement field clearly shows a major discontinuity that, starting from the southern slope of Mt. Vettore, rises to the western sides and exceeds, for many kilometers, to the rocky outcrop of Monteprata (Figure 4). The same ruptures reactivated following the main event in Norcia, together with a complex pattern of synthetic and antithetical faults, often subparallel but also intersecting each other.

The slope map (Figure 4) clearly highlights the continuous NNW–SSE trending alignment of the ruptures along the western flank of Mt. Vettore and the scattered fractures in the northernmost sector. The results related to the event of October 30 appear much noisier than those related to the Amatrice earthquake. This is due to the strong motion and numerous small local deformations, probably attributable to gravitational movements. It is possible to clearly identify the long break at the foot of Mt. Vettore, in the plain of Castelluccio di Norcia, responsible for road damages (see Section 4.2.2.). The ruptures along the crest of the Mt. Vettore massif are also well identified, as well as a small portion of the antithetic splay of Monteprata. Other possible ruptures, corresponding to known tectonic features, have been identified. Overall, the results match well with the data collected in the field during 2016 and 2017 geological surveys.

*4.2. UAV Mapping of the Mts. Sibillini Coseismic Ground Ruptures*

One of the main challenges in mapping coseismic ruptures in the Mt. Sibillini area is to identify and locate the path of the primary fractures. A complex system of faults which originated during the distinct tectonic phases have affected this region since the Mesozoic. After the main seismic events, in the last months of 2016, UAV mappings and field surveys were carried out for a few weeks, depending on the weather conditions. In areas higher than 2000 m a.s.l., the survey was completed in the spring/summer 2017.

Overall, 35 km of ruptures were detected from the Tronto river valley, in the south, to Muccia in the north (Figure 3). In the following, we report the description of two rupture zones in distinct portions of the Mt. Vettore area, both crossed by the paved road SP 477 (j, k in Figure 4); the first one is located on the southeastern slope, the other in the endoreic intermontane basin of Pian Grande (average altitude of 1300 m asl) in the western sector.

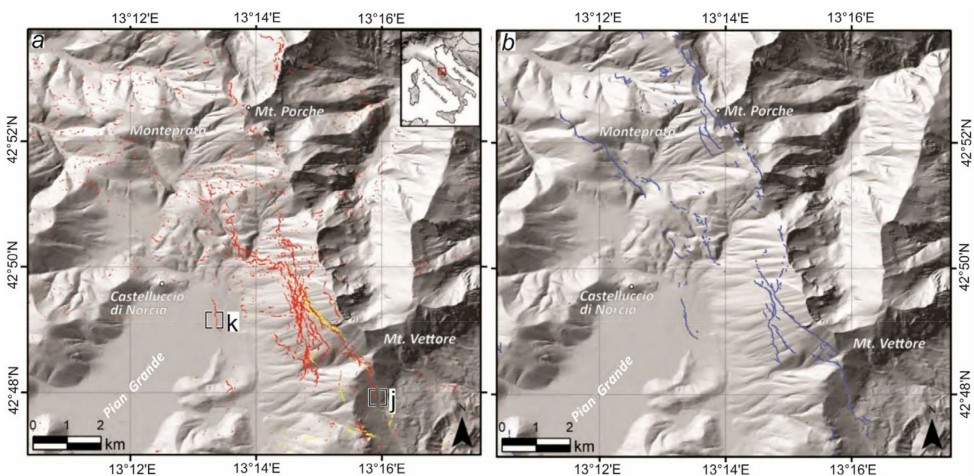

**Figure 4.** (**a**) Interferometric optimized slope map of the coseismic ground ruptures in Castelluccio di Norcia and Monte Vettore areas. The yellow lines are the gradient lineaments related to the Amatrice 24 August 2016 Mw 6.2 event, while the red ones are related to the Norcia Mw 6.5 event; the location of SE Slope of M.Vettore (j) and Pian Grande di Castelluccio di Norcia (k) are indicate; (**b**) Field map of the main coseismic ruptures related to both seismic events mapped during 2016 and 2017 field campaigns.

### 4.2.1. SE Slope of M.Vettore

Along the SE slope of Mt. Vettore, upstream of road SP 477 (Figure 5), the ruptures ran for a cumulative length of 1500 m, with an average N340° direction (Figure 5b), reaching the summit of the ridge and continuing on the western side of the mountain slope, bending to a N300° direction.

The ruptures consisted of open cracks and vertical dislocations or warps, with a few sinking structures (Figures 6 and 7). The cumulative displacement had an average offset along the slope (Sd) of 0.3 m and an average extension (Hd) of 0.14 m (Figure 8), mainly down-dip with a feeble left lateral component (Figure 5c). The ruptures involved the soil, the debris slope deposits and the carbonate substratum. Minor slip and shaking effects also took place along nearby synthetic and antithetic normal faults distributed along the slope. The distribution of the displacement (i.e., the free face scarp) along the rupture trace (Figure 8) showed the segmentation of the main fault segments, with one central displacement peak and at least five other secondary peaks separated by minima corresponding to bends and stepover sectors. The rupture extension (Hd) along the fault trace was complex, and a direct correlation with the slope offset (Sd) was not observable.

The UAV survey carried out on 31 October 2016 (Figure 5) partially covered the rupture for a length of about 500 m and was focused in the northern and highest sector where xerophilous grassland characterizes the mountain slopes and the surface deformation is not hidden by the vegetation. A detailed DSM was produced with a cell resolution of about 0.01 m/px (Figure 5), along a slope where the inclination toward ESE is higher than 22°/25°. In the lower part, the ruptures consisted of several arrays affecting the bare ground, crossing the asphalt road pavement of the SP 477 in two sets with an offset of 0.12–0.15 m and aperture of a few cm (Figure 7) [80]. These fractures corresponded to a normal fault alignment dipping to the WSW at about 60° (Figure 5) that juxtaposed the Upper Jurassic cherty limestone of the hanging wall to Middle-Low Jurassic limestone of the footwall, with a geological throw of a few hundred meters. It is important to note that the downslope ruptures in the road pavements were also visible in the image of May 2011 from Google Street view [81] (Figure 7a). Their origin could be related to the upward swelling (frost heave effect) of water along preexisting fractures. These asphalt cracks seem to have been reactivated by normal faulting during the 25 August 2016 event, with an offset of a few cm. In September 2016 the road ruptures were restored, but the 30 October 2016 event broke the new asphalt, partially overprinting the same cracks (Figure 7c).

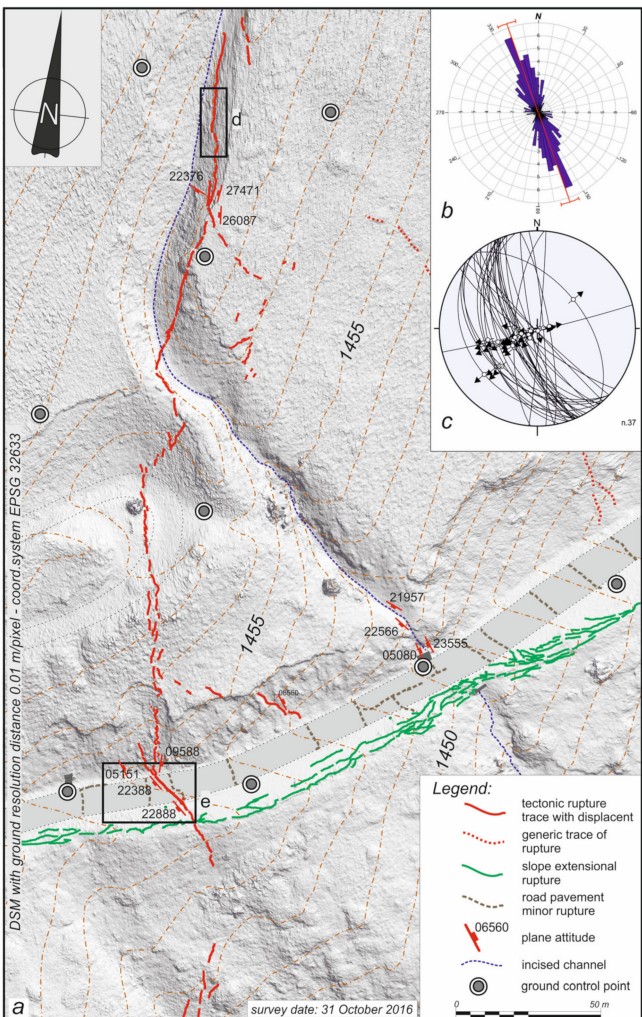

**Figure 5.** (**a**) DSM of the SE Slope of Mt. Vettore, above the road SP 477 (see location in Figure 4), obtained from UAV survey carried out 31 October 2016, with a cell resolution of about 0.01 m/pixel; location of Figures 6 and 7 are indicated. (**b**) Rose diagram of the frequency distribution of the rupture azimuth (the red line is the mean value; the arc is the mean error distribution for the 95% confidence interval); and (**c**) Schmidt lower hemisphere projection of the fault planes and slip vectors; (**d**) Location of Figure 6; (**e**) Location of Figure 7.

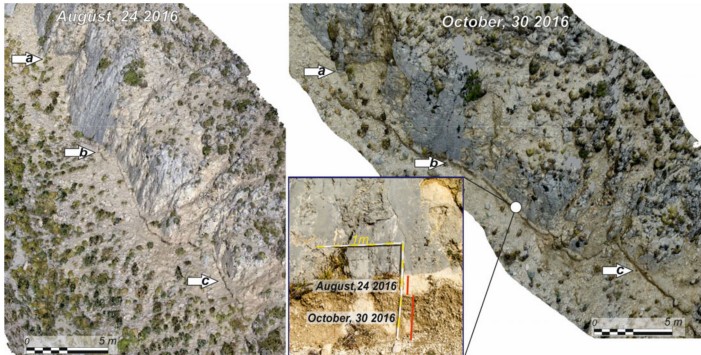

**Figure 6.** The 3D point cloud oblique view of the SE slope of Mt. Vettore (location in Figure 4), with the coseismic ruptures of the 24 August and 30 October 2016 events. For comparison, the same sites (**a**–**c**) are indicated in both images. In the center, a detail showing the displacement relative to the two seismic events.

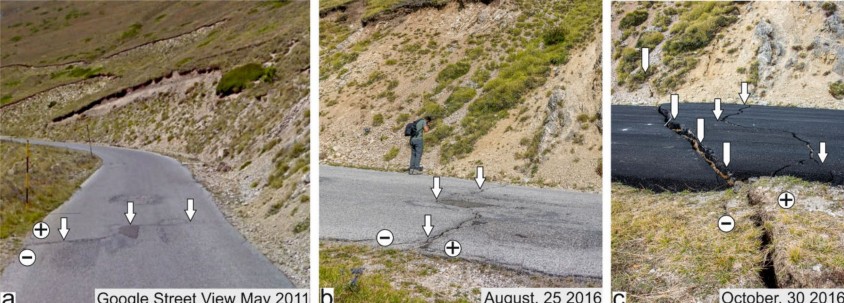

**Figure 7.** Multitemporal view of the coseismic ruptures in the asphalt pavement of the Strada Provinciale n. 477 on the southern slope of Mt. Vettore (see location in Figure 5); white arrows indicate the same location at different times; (**a**) Google Street View from May 2011 [81], where a small down-slope step is observable; (**b**) oblique image of the coseismic ruptures after 24 August 2016 event, during which the previous discontinuity was reactivated in the upslope; (**c**) oblique photo of the coseismic ruptures after the 30 October 2016 event, during which the reactivation of the previous fractures occurred and new fractures were generated (white arrows).

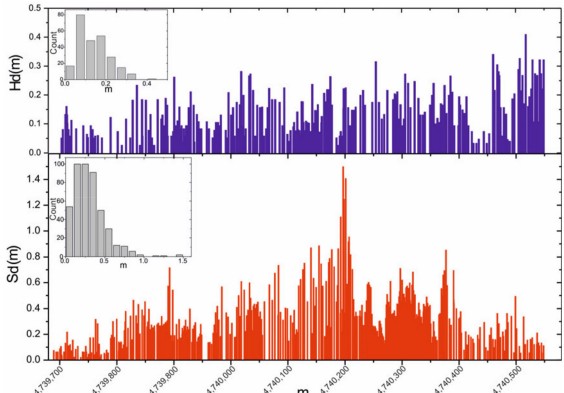

**Figure 8.** Horizontal (Hd) and slope (Sd) total displacement of coseismic ruptures on the southeastern slope of Mt. Vettore, plotted as a function of UTM Northing (m). The visible displacement peaks correspond to the different rupture strands. Inset graphs are the frequency distribution of the displacements.

### 4.2.2. Pian Grande di Castelluccio di Norcia

The Pian Grande di Castelluccio di Norcia plain is an intramountain basin in the core of the Mts. Sibillini, with endoreic drainage towards the south where the karstic sinks are located [82]. It consists of a flat grassland at an average altitude of 1300 m, used for sheep-farming and partially for lentil cultivation. The plateau has a rhomboidal shape of about 15 km², with the main axis oriented N–S, a length of 6 km and a width of 2.8 km (Figure 4a). The plain is enclosed by two ridges with steep slopes; Mt. Vettore to the east, reaching an altitude of 2400 m a.s.l., and minor mountains to the west, reaching altitudes of 1800 m a.s.l. The basin developed in the hanging wall of the SW-dipping normal fault-system [83], with cumulative throw, distributed at least in three fault segments, of more than 1000 m. The faulted blocks in the hanging wall consist of Lower Cretaceous limestone, while Lower-Middle Jurassic limestone crops out in the footwall, both pertaining to the Umbria-Marche succession [84]. The basin is filled by sequences of Late Pleistocene–Holocene continental fluvio-palustrine and fluvio-glacial sediments, including Last Glacial Maximum (LGM) sequences [85]. These sediments are interfingered with debris and alluvial fan sediments, fed from the slope where highly fractured limestone crops out [82]. A geophysical survey and a few shallow boreholes indicate a complex stratigraphy, with three main basin depocenters that reach a maximum depth of 300 m [86–88].

During the 30 October 2016 event, three sets of coseismic ruptures formed in the Pian Grande: one in the eastern sector, with a 1500 m long west-dipping free face, parallel to the Mt.

Vettore–M.Bove fault system (Figure 4a), and two other ground fracture systems which were located south of the Castelluccio village, with the downthrown block to the east, showing lengths of 1100 m and 200 m (Figure 4). All the ground ruptures developed in a sector of the plain characterized by a ridge and furrow system with a drainage groove, where crops grow on soil formed over large alluvial fan deposits of coarse gravels with a matrix of sand and silt. The coseismic fractures intercept the site where a paleoseismological [87] trench has been dug, allowing the identification of three displacement events (4 k/6.8 k/15 k years BP) [89]. Many ruptures, also visible in INSAR slope maps, were well defined, especially the easternmost one, which intercepts and breaks the paving of the road SP144 in several points (Figure 9a,d,g,h). On the low altitude orthophotos acquired with UAVs, and in the DSM produced with a ground resolution of 0.02 m/pixel, the rupture pattern was characterized by a series of N–S striking (Figure 9e) en-echelon-arranged ruptures with an offset of many centimeters, and the western blocks were downthrown.

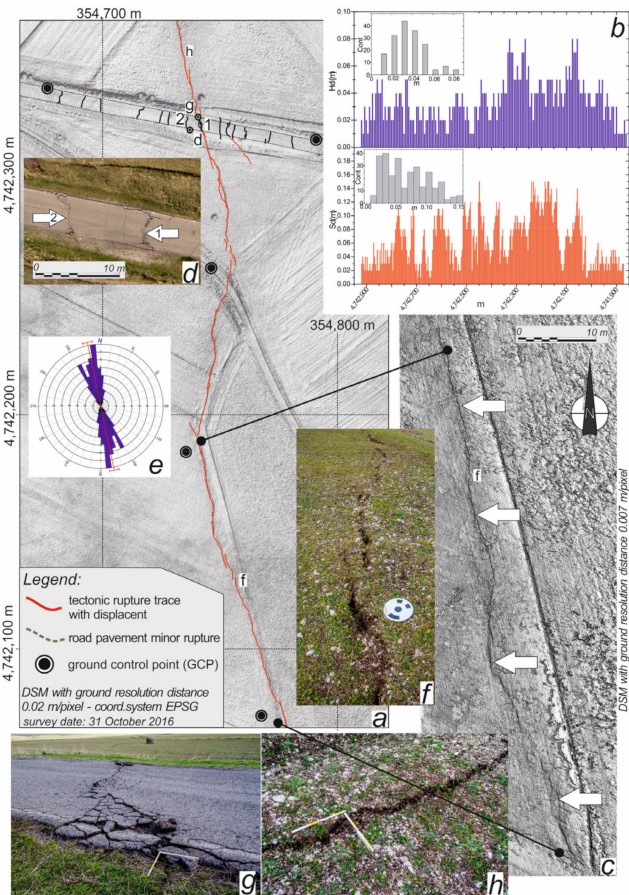

**Figure 9.** (**a**) DSM of Pian Grande di Castelluccio di Norcia (see location in Figure 4) obtained from UAV survey carried out on 31 October 2016, with a 0.02 m/pixel resolution; the rectilinear features correspond to the agricultural crops; location of the photos d, f, g, h is indicated. (**b**) Horizontal (Hd) and slope (Sd) displacement of coseismic ruptures, plotted as a function of UTM Northing; inset graphs are the frequency distribution of the displacements. (**c**) DSM with 0.007 m/pix resolution showing the coseismic ruptures in the field. (**d**) Oblique aerial view of the ruptures 1 and 2 (white arrows) indicate in (**a**) (**e**); Rose diagram of the frequency distribution of the rupture azimuth (the red line is the mean value; the arc is the mean error distribution for the 95% confidence interval). (**f**) Detail of the right en échelon fractures pattern—the size of the disk is 20 cm. (**g**) Fractures in the road pavement—the rule is 40 × 40 cm. (**h**) detail of the ground rupture—the rule is 40 × 40 cm.

The offset was measured in detail at about 300 localities along the entire length of the surface rupture in November 2016 (Figure 9b). The ruptures were detectable in the DSM at a very high resolution of 0.007 m/pixel (Figure 9c), despite the agricultural cultivation (Figure 9f,h). In the spring 2017, all the coseismic ruptures were completely erased by the cultivated fields and by the new asphalt pavement along the road. The average vertical displacement (Sd) over the full length of ground ruptures was ~0.06 m (Figure 9b), while the horizontal displacement (Hd) was 0.03 m, distributed across a ~0.50 to 5 m wide deformation zone. On the road pavement, the offset showed a small amount of shear, and the asphalt fractures were distributed for a length of more than 100 m (Figure 9g). The distributed pattern of deformation reflected a considerable thickness of poorly consolidated alluvial gravel deposits. The distribution of these ruptures (Figure 9a) was approximately symmetrical along the ~1.5 km fault strand, until the rupture tip zones where overall displacement was less than ~0.01 m. In the southernmost section, the net displacement was >0.10 m, with three displacement peaks and at least seven other secondary peaks separated by minima corresponding with bends and stepover in the fault trace (Figure 9b).

### 4.3. Mt. Etna Interferogram Analysis

For the Mt. Etna area, two interferometric pairs of Sentinel-1 IW single look complex (SLC) images from descending track 124 and ascending track 44 were processed (Figure 10). The acquisitions cover the pre- and post-seismic periods, from the 22 to the 28 December 2018.

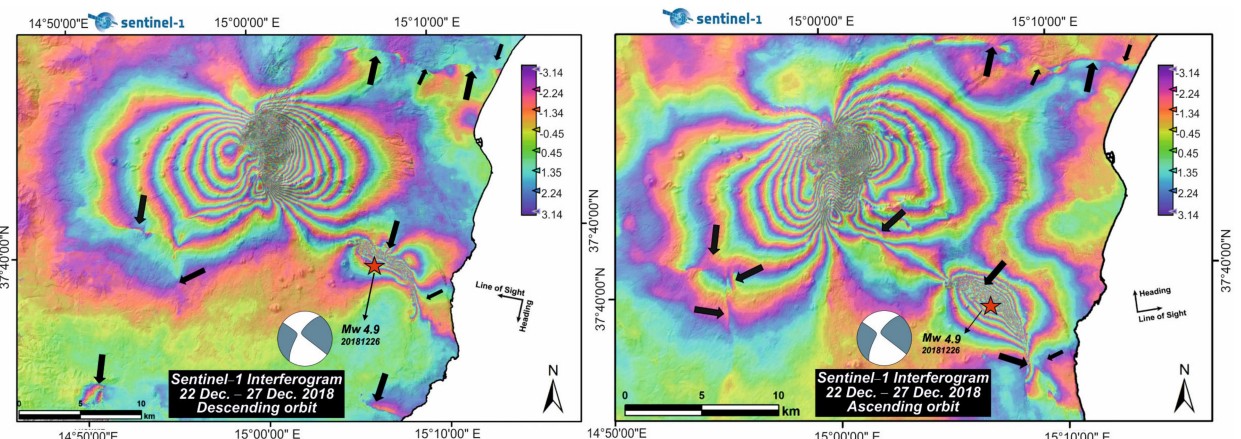

**Figure 10.** Sentinel-1 interferograms showing the effect of the 26 December 2018 earthquake at Mt. Etna. The red star indicates the epicenter of the earthquake; arrows indicate the most evident effects.

The interferograms were flattened with a flat-earth phase algorithm, and the topographic phase was removed based on the about 30 m (1 Arc-Second) reference DTM (SRTM). To reduce the effects of phase noise, Goldstein adaptive filtering [90] was applied (Figure 11). Phase unwrapping was successfully performed using SHAPHU [91] (Statistical-Cost, Network-Flow Algorithm for Phase Unwrapping), optimizing parameters to avoid unnecessary displacement field smoothing. The Sentinel-1 data permitted the reconstruction of the coseismic deformation field, with fringes' patterns characterized by two asymmetric lobes and a four-symmetrical quadrant distribution pattern, which is typical of earthquakes with a strike-slip mechanism (Figure 10). In detail, the fringe patterns showed a small asymmetry with a positive LOS displacement ascending deformation field in the western side, indicating that the seismogenetic fault dips towards the east.

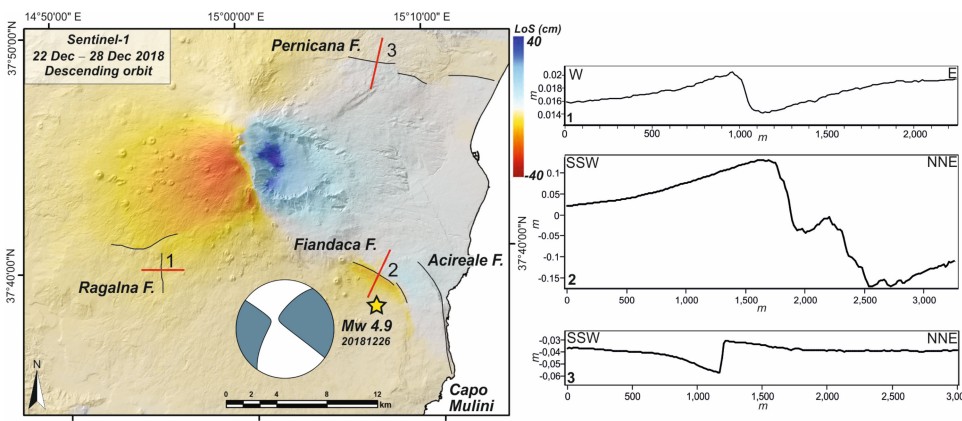

**Figure 11.** The deformation field map obtained from the unwrapping of the Sentinel-1 descending orbit interferogram; epicenter (yellow star) and focal mechanism of the 28 December 2018 are indicate (**left**). The differential displacement along the major fault systems is highlighted in the cross sections for Ragalna (1), Fiandaca (2) and Pernicana (3) faults (**right**).

The results were geocoded for interpretation and field checking. The interferometric processing highlighted, with good detail, a complex deformation field that affected the entire volcanic edifice (Figure 10). The summit area was affected by displacement due to magma ascent, reaching more than 40 cm and 65 cm along W and E directions [76].

The coseismic slip of the Fiandaca fault resulted in a predominantly horizontal movement of 15 cm towards the E and 20 cm towards the W (Figure 12). Small-scale deformation patterns were also detected along other faults of the eastern and southern flanks of the volcano, where aseismic slip occurred inside the timespan of the interferogram. Thanks to the high final coherence achieved, many discontinuities were easily detectable from the wrapped phase. We tested various approaches to enhance their assessment; linear loss-of-coherence areas were examined, corresponding to fault lineaments. Moreover, an edge detector filter was applied to the unwrapped phase, analyzing the displacement field like a morphological feature [54]. This approach is very efficient even though it is susceptible to phase unwrapping errors. Finally, a phase gradient computation was conducted, directly from the real and imaginary parts of the interferograms using the expression by Price and Sandwell [35]. The result highlighted the complex pattern of ground ruptures along the Fiandaca fault, including its southern termination, where strong decorrelation was found due to the coseismic movement (Figure 12).

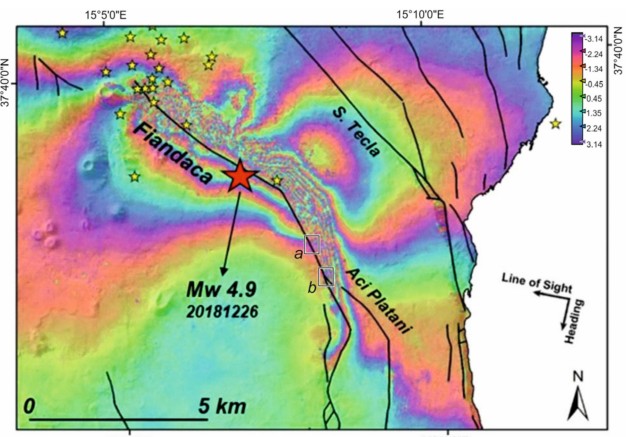

**Figure 12.** Detail of the Sentinel-1 descending orbit interferogram centered on the Fiandaca fault. The hypocenter of the main earthquake of 26 December 2018 (red star) and the aftershocks (small yellow stars) are shown [54]. Rectangular frames are the locations of (**a**) Figure 13 and (**b**) Figure 14.

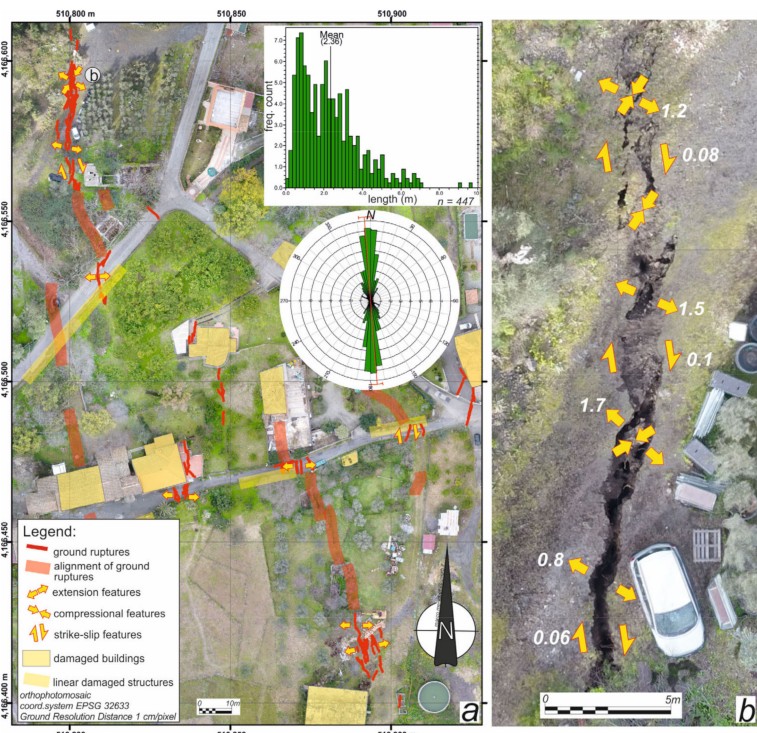

**Figure 13.** (**a**) Orthophotomosaic of the Pennisi area (location in Figure 12), obtained from the UAV survey of 14 March 2019 with a 0.01 m/px resolution, showing coseismic ground ruptures and damaged buildings; insets are the histogram of the frequency distribution of rupture lengths in meters and the rose diagram of the frequency distribution of their azimuth (the red line is the mean value, the arc is the mean error distribution for the 95% confidence interval); and (**b**) detailed view of the point b, showing a large fissure with compressional and extensional features related to right-lateral strike-slip motion (numbers refer to extension in meters).

Linear features have been also detected along other major faults crossing the volcano. In particular, the interferometric data recorded in good detail the activity of the Pernicana and Ragalna faults, which have undergone slow aseismic movements during the time span of the acquisitions (Figure 11). They presented such a good resolution and spatial continuity that they could easily be used to improve geological maps. In fact, along these linear features, fresh ruptures have been found, especially along the roads, even in the presence of hidden tectonic structures such as the one crossing Capo Mulini at the southern termination of the Acireale Fault (Figure 11).

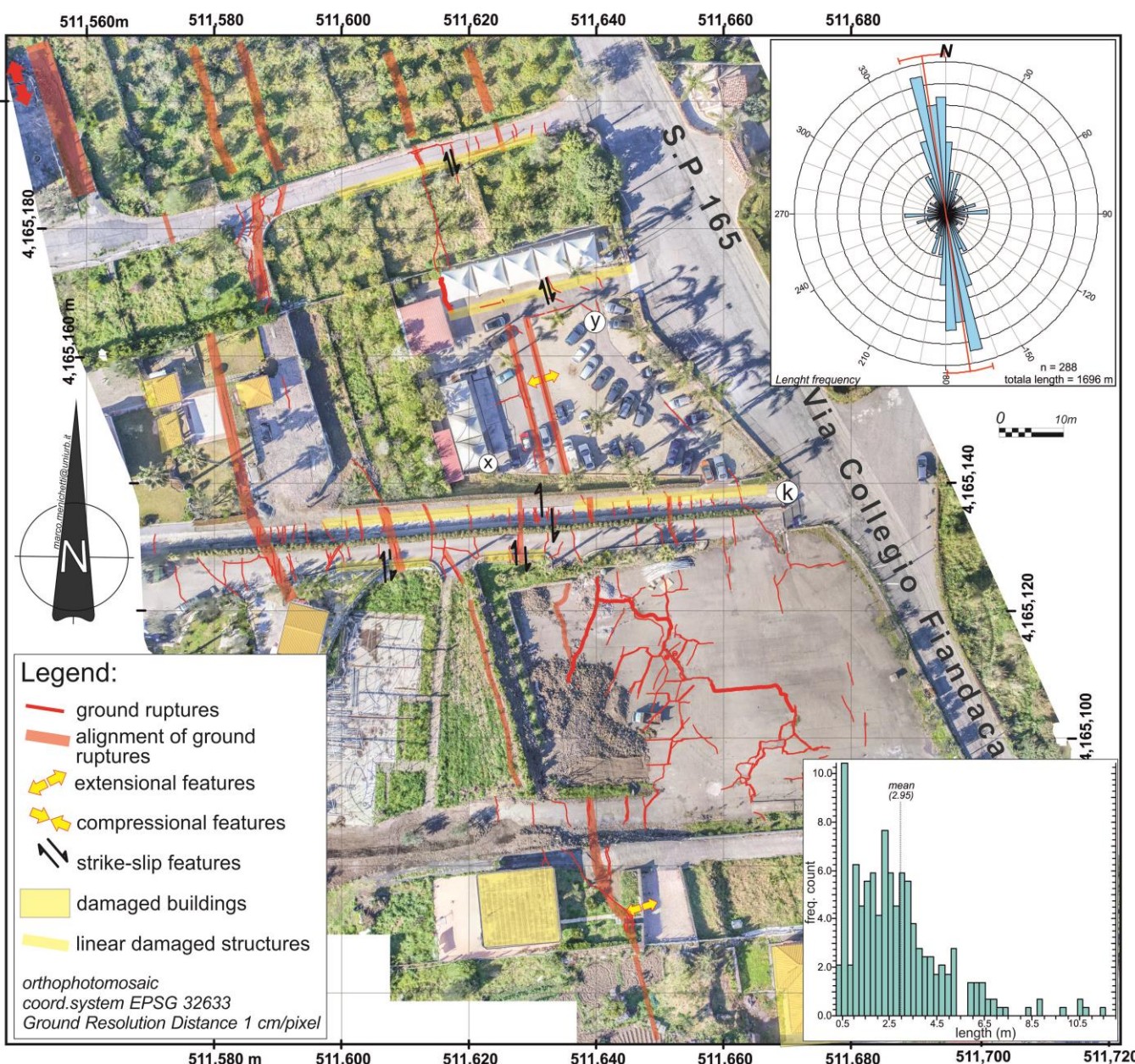

**Figure 14.** Orthophotomosaic of the Fiandaca area (location in Figure 12) obtained from a UAV survey of 14 March 2019, with a 0.01 m/pix resolution; coseismic ruptures are visible in the ground, paved roads and walls; insets are the rose diagram of the frequency distribution of the azimuth of the fractures (the red line is the mean value, the arc is the mean error distribution for the 95% confidence interval) and the histogram of the frequency distribution of length ruptures in meters. Location of reference points (x, y, k) of Figure 15 is indicated.

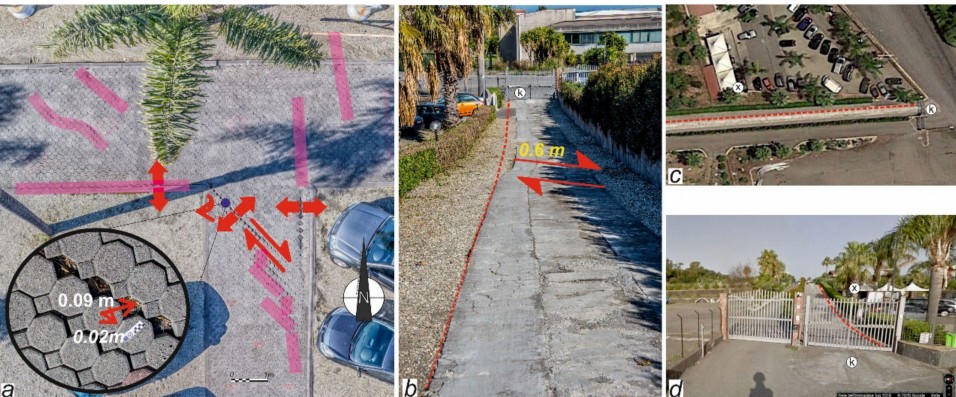

**Figure 15.** Ground ruptures formed during the 26 December 2018 seismic event along the Fiandaca fault (red arrows indicate the movement direction-cyan strips are the deformed zone-dashed red line indicate the undeformed lane edge): (**a**) displacement of interlocking bricks in a parking pavement (point y in Figure 14) associated with right lateral strike-slip motion; (**b**) displacement of a private lane surveyed in March 2019 (from point x to point k in Figure 14); (**c**) the same area without visible ground deformations in a Google Earth image of 21 July 2018 [92]; and (**d**) Google Earth Street View from point k (July 2018) [93].

### 4.4. UAV Mapping of Mt. Etna Coseismic Ground Ruptures

The investigated area is located on the southeastern flank of the volcano at an average altitude of 450 m, and it is characterized by a gentle slope. Here, anthropic activity has greatly changed the landscape and the soil is mainly used for agricultural purposes. The substratum consists of basaltic lava and, at a few points, retaining walls and terraces delimit gardens. The area is crossed by a complex network of dirt roads, some with asphalt pavement, useful for recognizing the geometry of coseismic ruptures.

After the earthquake of 26 December 2018, in the first weeks of 2019, we mapped the coseismic ruptures before they were restored, especially in the road pavements and in the courtyards of the houses. At the same time, UAV surveys made it possible to acquire low altitude zenithal and oblique aerial photos of key sectors of the reactivated structures. The produced orthophoto maps, with a ground resolution of 0.01 m/px, allowed us to discern some features of the complex pattern of coseismic fractures. Loose soil and vegetation cover hid the ruptures, and many of them were only detected by direct observation in the field. However, many of the detected ground ruptures were also visible in the interferogram slope maps (Figure 11). Sets of ruptures for more than 13 km were detected in an arcuate alignment extending from the village of Fleri, in the northwest, to Aci Trezza in the southeast, largely corresponding to the Fiandaca fault (Figure 1). Below is a detailed survey of two key areas, the first located a few hundred meters southwest of the village of Pennisi and the other, to the south, at the site of Fiandaca (Figure 12). The geology of both sites consists of a basaltic lava flow and pyroclastics, attributed to the 1326 B.P. eruption [53], locally covered by alluvial and colluvial deposits. Moreover, to verify the proposed methodology, we surveyed two other structures (the Ragalna and Acireale faults) along which the interferometric slope maps highlighted lineaments that could be interpreted as ground ruptures related to the event of 26 December 2018, even though their origin is quite controversial (see Section 5).

#### 4.4.1. The Fiandaca Fault in the Pennisi Area

In this area, 447 coseismic fractures have been recognized in the orthophotos for a length of about 1 km (Figure 13). They consisted of parallel strands and, most frequently, of left-stepped en echelon open cracks with different lengths, spanning from a few dm up to 10 m (average length of 2.3 m, see the histogram in Figure 13). These subvertical fractures were distributed along a 5 m up to 50 m wide strip. They were oriented mainly N350°, with low dispersion values of the azimuthal data (see the rose diagram in Figure 13).

In the northern strand, a 1–2 m deep ground crack, 50 m long and with an extension variable from 0.5 up to 1.7 m, was observed (Figure 13b). This fracture presented a complex kinematic with prevailing pull-apart geometries (average extension less than 1 m) related to a right-lateral strike-slip motion of about 0.1 m and a resultant slip vector oriented ESE. To the south, smaller subparallel fractures were visible over a road pavement, damaged buildings, and other man-made structures (Figure 13a). During the earthquake, stone walls collapsed and several masonry constructions suffered serious damage, with fractures and rotations of the brick walls. Man-made linear markers, such walls and paved roads, made it possible to reconstruct the fracture traces, even though the areal distribution and the length of the identified rupture was underestimated. It is worth noting that during the first months of the 2019 the offset increased due to aseismic creep.

### 4.4.2. The Fiandaca Fault in the Fiandaca Area

The Fiandaca area is an urbanized zone where several roads of many private properties and infrastructures meet (Figure 14). The UAV survey was made in March 2019, a few months after the main seismic event of 26 December 2018, during the aftershock swarm. In this area, 288 coseismic fractures were recognized, showing a prevalent N340° trend (Figure 14). Many ruptures deformed sidewalks, curbs and pavements, with complex patterns such as dilatational jogs and block rotations linked to soil deformation (Figure 15). Generally, the fractures were organized in subparallel strips, about 100 m wide, overlapping with sinistral en echelon arrangements. Small east–west trending reverse faults were also observed in the overlapping sectors between en echelon right-lateral strike-slip segments. The average length of the strands was about 3 m over a total length of more than 1 km (Figure 14). In this area, coseismic displacement was observed in a car dealer's parking area, where the ground was partially paved with asphalt and bricks (Figure 15a). These interlocking bricks allowed the reconstruction of the cumulative deformation, characterized by an extension of 0.2–0.5 m and a right-lateral strike-slip motion of 0.1–0.3 m. A right-lateral strike-slip displacement of about 0.6 m was also measured on the pavement of a private lane (Figure 15b). The same road without ruptures is visible in the Google Earth image of 21 July 2018 [92] (Figure 15c) and in the Google Earth Street View [93] taken in July 2018 (Figure 15d). To the south, in the asphalt pavement of a forecourt (Figure 14), the offset was distributed in a complex network of fractures, hundred of meters long, in property division walls, garden roads and damaged buildings. These measures were affected by aseismic creep deformation many weeks after the main seismic event, and 25% further extension was measured the following month [24].

### 4.4.3. The Ragalna Fault

This fault extends along the southwestern slope of M. Etna, where a N–S oriented morphostructural lineament, reported in several geological maps as the Ragalna Fault [94], is clearly visible over the DEM. It is a trastensional right-lateral subvertical (east-dipping) fault that, in the northern sector, shows a 5 km long and up to 20 m high fresh scarp that developed on 15 Ka-old volcanics. At its southern end, this lineament is characterized by the occurrence of cinder cones aligned in a N–S direction. To the south, the fault loses its morphologic evidence, and a 3 km long fracture zone develops with a NNE–SSW direction towards the village of Santa Maria di Licodia [95].

The Ragalna Fault is evident in the phase gradient map, and in an E–W section the total displacement reached a few cm (diagram 1 in Figure 11). The area is strongly cultivated and modified by the agricultural activity, and the ground ruptures formed after the 26 December 2018 event are visible only in a country road pavement and on the adjacent wall as a set of small NE–SW striking fractures with left-stepping en echelon arrangement (Figure 16).

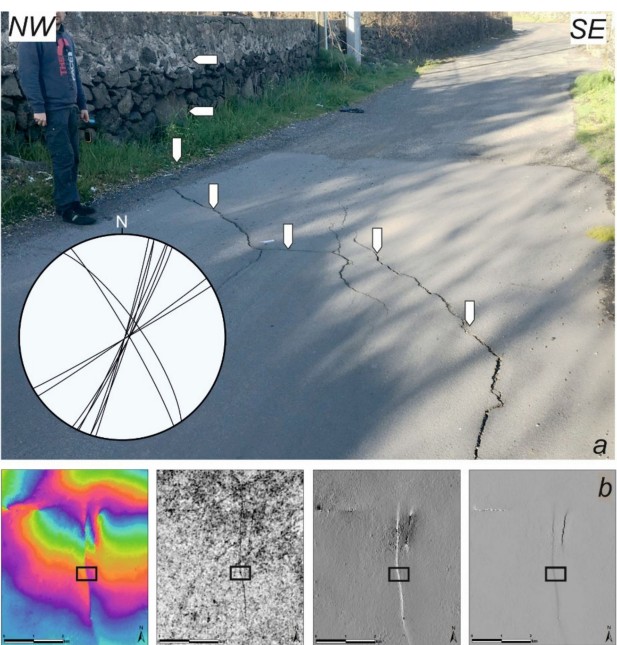

**Figure 16.** (**a**) Centimetric ground ruptures in an asphalt road pavement and in a dry stone wall (white arrows) along the Ragalna Fault (see Figure 11—coordinates 37°39′23.1″N–14°55′15.3″E). The attitude of the fracture planes is shown in the Schmidt lower hemisphere stereogram. (**b**) Linear features are also visible in the Sentinel-1 InSar interferograms analysis; from left to right: phase, coherence, phase gradient and slope maps (square indicate le location of photo (**a**)).

### 4.4.4. The Acireale Fault

The Acireale fault forms a 7 km long, NNW–SSW striking, rectilinear scarp along the Ionian coast, where it forms a cliff more than 150 m high (Figure 1b) on a volcanic sequence mostly constituting 200 to 100 Ka-old rocks [40]. At the southern end of this structure, in the Capo Mulini site (Figure 17), it forms a 10 m high scarp, delimiting a narrow terrace for more than 1 km parallel to the coast. The phase gradient map shows a N–S striking lineament that reaches the total displacement of a few cm (Figure 11). The survey in this area permitted the recognition of a few ground ruptures in a road pavement and on the adjacent wall as a set of small N–S striking fractures. These ruptures are not visible in the Google Street view of July 2018 [93].

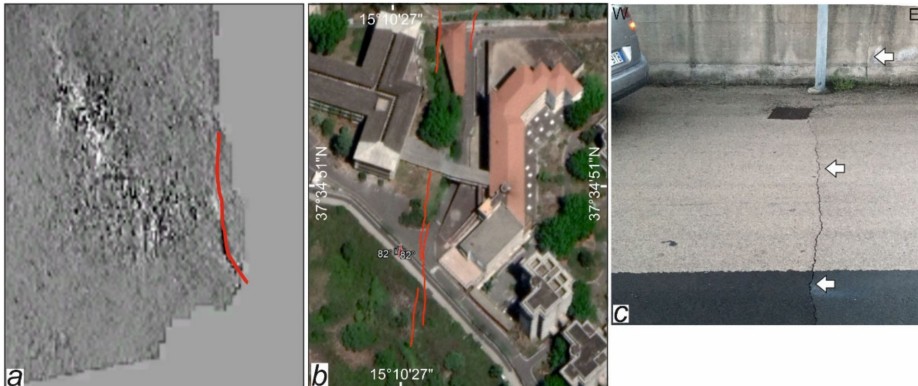

**Figure 17.** (**a**) Phase gradient ascending Sentinel-1 interferogram highlighting the southern end of the Acireale Fault (red line). (**b**) Google Earth image of the surveyed ground ruptures (red lines) in the Capo Mulini site (see Figure 11—coordinates: 37°34′50.1″N–15°10′28.1″E). (**c**) Road pavement asphalt with centimetric ground ruptures (photo March 2019).

## 5. Discussion

The survey of coseismic ground ruptures is central to define kinematics and dynamics of active faults and to assess the seismic hazard of a region. Earthquake events that produce coseismic surface deformation generate offset in the landforms related to the magnitude and kinematics of the seismic event. Generally, measurements of the displacements across surface ruptures in different tectonic scenarios are performed on single fault segments and collected by a local field survey, often with the aid of GNSS receivers.

The ground survey of coseismic ruptures on large regions is long, time-consuming hard work, in all the mophostructural contexts including steep rugged mountain slopes, vast desert areas and complex urbanized regions [2]. In recent years, the aid of spaceborne Lidar and photogrammetry, with sub-meter scale sampling, has allowed the surveying of large areas affected by coseismic ruptures [21]. On the other hand, the availability accomplishment of an airborne survey needs to be planned over time, sometimes months or years after the earthquake, with consequent wear and loss of data.

High-resolution mapping based on UAV represents an opportunity to gather three-dimensional high-resolution topographic datasets. The centimeter-resolution images obtained by SfM photogrammetry allow the details of the vertical and horizontal displacement to be mapped in 3D, and constrain the spatial geometric characteristics of active faults. The analysis of coseismic ruptures in tectonic and volcanic contexts indicates that about 40% of the total surface displacement occurs as off-fault deformation, over a mean deformation width of a few hundred meters. In similar tectonic setting, normal faults have determined distributed deformation with even higher percentages (about 66%) and considerable rupture zone widths [95]. The rupture zone fabric and the off-fault deformation is mostly controlled by the structural complexity of the fault system, with a weaker correlation with the rheology of the rupture materials [5].

The proposed method permits a preliminary and rapid reconnaissance in the field of geometries and an offset of coseismic ground ruptures generated by strong earthquakes. The combination of aerial and terrestrial surveys allows the discrimination of primary and secondary surface faulting (directly connected to the earthquake source), associated ruptures and superficial and deep gravitational phenomena.

The interferometric analysis results in a powerful method to preliminarily localize the area in which to focalize detailed surveys of coseismic ruptures. In this work, we tried to validate the analysis of InSAR products for the detection of linear ground displacements. For this purpose, two different real case studies are presented, in the Mts. Sibillini and Mt. Etna areas. In these areas, coseismic surface ruptures occurred along pre-existing structures and a preliminary survey [46,47] showed the general pattern of the ruptures across regions that covered more than 400 km$^2$. From a methodological perspective, exploiting the SAR data allows a quickly preliminary map of the possible ground ruptures and relative branches to be obtained. The results demonstrated the effectiveness of the C-band Sentinel-1 open data for this purpose. These approaches could also allow for the detection of the propagation of cracks during a seismic sequence characterized by many major events. In addition, they can be useful as a field guide to speed up detailed survey, especially in large or remote areas. The joint application of several techniques allows us to extrapolate as much information as possible from the interferometric data. However, there are many limits, for example in very vegetated places, where detecting ground ruptures is not possible. In addition to this, the limited resolution of the radar data makes it possible to identify only the areas that may have undergone deformations, not the actual fractures. Often, due to unwrapping errors and other processing reasons, false positives are present. The results must be well interpreted by an expert. Improved results could be achieved using a high resolution DTM for topographic phase removal and defining good strategies for filtering and classifying results.

The maps of the coseismic ruptures reported in the preliminary surveys of a sector of the Mt. Vettoretto flank (e.g., Figure 3 of [46]) show some significant differences with respect to our Figure 5. The spatial distribution and the accuracy of measurement of the offset

of the DSM is the highest by at least of one order of magnitude [44,46,47]. Additionally, the directional data of the fractures in bedrocks and in debris and soil have a difference of 20–30 degrees, values than cannot be negligible in a structural analysis. Moreover, the high-resolution DSM of the area permits the verification of the control of the local topography on the fracture pattern along the slopes in the correspondence of stream channels or scarps, where the gravitational contribution could be significant. The available database on the trace of coseismic ruptures in the Mts. Sibillini area reports observations collected in about 7000 points [47]. Comparing the interferometric slope map of this area (Figure 4) with the field survey data, we noted a significant increase in the geometric and structural data of coseismic fractures of over 25% in the database [78]. The regional distribution of deformations shown by interferometric maps can be extended many kilometers off the surface ruptures, making an important difference in the evaluation of the empirical all-slip-type relationships between magnitude and surface rupture length/average surface displacement [77]. In a recent paper [96], the coseismic deformation pattern of the Mts. Sibillini 2016 earthquakes, extracted by DInSAR techniques from ALOS-2SAR data, identified a large set of surfaces ruptures like those shown in our Figure 4. In general, the proposed methodology, compared with more detailed structural analyses of the coseismic ruptures [25,40,80], improved the data quality.

In the Mt. Etna area, the distribution and geometry of coseismic fractures related to the 26 December 2018 earthquake was strongly affected by the lithology of the volcanic products [49,74,81,82]. Here, it was fundamental the role of the anthropic artifacts for the reconstruction of the fracture pattern, even though diffraction effects in the direction distribution were observed along the whole fault strand [82]. The indications gathered from the interferogram permit the highlighting of the areal distribution around the volcano structures where the far field deformations of the active faults are difficult to recognize. Several descriptions of the coseismic ruptures have been already published, with different scales of details related to different purposes. The preliminary survey of Civico et al. [79] is synthetized in a general map of the area where the distribution surface ruptures, and is focused on the main noticeable structures. Further published data [69,72,81,83,84] added many other details. In our survey, the brittle fractures were mainly localized on anthropic manufactured aspects (concrete and asphalt) where the mechanical proprieties permit a better preservation. Large deformations, with offsets larger than meters, can be observed only where the soil is thicker. The acquired aerial low altitude images in the Pennisi and Fiandaca areas allowed the geometry of the deformation zone to be detailed, to distinguish the different generations of ruptures and to determine their spatial distribution. The areal distribution of coseismic ruptures was detected over a radius of less than one hundred meters, according to the other published data [72,85]. The interferometric map also showed the reactivation of the Ragalna Fault, in the southern slope of the volcano, and of the southernmost sector of the Acireale Fault along the Ionian coast. In our opinion, the main seismic event shook many other tectonic structures, including geological contacts and discontinuities, triggering their displacement as aseismic creep.

Our approach, including interferometric analysis and field survey, adds new information for producing a more complete picture of ground deformations. This is an important consequence in terms of the definition of areas involved in fault deformation and in terms of seismic hazard assessment. Precise and more accurate measurements of displacement along coseismic ruptures can be acquired using recently available remote sensing platforms.

## 6. Conclusions

The recognition and structural analysis of surface coseismic ruptures are essential to characterize seismogenic structures and improve regional seismic hazard assessments. Unprecedented interdisciplinary observations have combined methods and models to help to advance the characterization of coseismic rupture geometries and allow 3D spatio-temporal surface displacements to be achieved.

The combination of InSAR areal investigation and a detailed UAV survey with geological field survey measurements locally validated and calibrated the SAR results. This is a robust technique used to characterize active faults that provides the best sampling in morphological complex regions where the data are very sensitive. It enables a highly detailed analysis of coseismic rupture geometries and allows not only for the measurement of offsets, but also the quantification of uncertainties more precisely and rigorously than was possible in the past. The compilation of offset measurements from coseismic ruptures poses few methodological problems because the different methodology used by the researchers inhibited the possible comparison and analysis of the results. Combining field work with remote sensing analysis allows the evaluation of the reliability of measurements and the quantitative constraints of uncertainty.

Sentinel-1 A/B Interferometric Wide (IW) Swath TOPSAR mode offers the possibility of acquiring images with a short revisit time. This huge amount of open data is extremely useful for geohazards monitoring, such as for earthquakes. Interferograms show the deformation field caused by the principal fault movements. Phase discontinuities appearing on wrapped interferograms, or loss-of-coherence areas could represent small ground displacements. However, phase gradient maps from complex interferograms could also be exploited to visualize lineaments.

In this work, a comparison between different methods has been made together with a field validation of results. We presented two case studies. The first one was relative to the 2016 central Italy earthquakes, astride which the InSAR outcomes highlighted quite accurately the field displacement of extensional faults in the Mt. Vettore–M. Bove area. Here, the geological effect of the earthquake was represented by more than 35 km of ground ruptures strands, with a complex pattern composed of subparallel and overlapping synthetic and antithetic fault splays. The interferometric slope map and the field survey of coseismic fractures in this area showed a significant increase, by more than 25%, of geometrical and structural data, with a directional difference of 20–30 degrees. The DSM permitted the detailing of the spatial distribution and the geometries of the coseismic ruptures, which was at least one order of magnitude larger with respect to the only field recognition.

The second case was relative to the Mt. Etna earthquake of 26 December 2018, following which several ground ruptures were detected. The analysis of the unwrapped phase and the application of edge detector filtering and other discontinuity enhancers allowed us to identify a complex pattern of ground ruptures. Here, with respect to the preliminary survey synthetized in a general map of the whole area, our approach showed much more detail of the surface ruptures, and better enlightened the deformation zone, especially in the urbanized area. In the Pennisi and Fiandaca areas, our approach permitted us to distinguish the different generations of ruptures and to determine a larger spatial distribution. The InSAR data permitted the identification of other, previously unknown ruptures pertaining to the Acireale and Ragalna faults.

The results suggest the need for an integrated approach combining remote sensing techniques and field survey, and also the proposition of models of fault kinematics in complex areas.

The proposed methodology sheds new light on coseismic displacement fields and ruptures' spatial geometries and fault kinematics, and on how different structural zones respond to coseismic surface displacements. Beyond that, our systematic chains of analysis can also be applied in full or in part to help understand other geodynamic contexts and heterogeneous tectonic environments.

**Author Contributions:** Conceptualization, M.M., M.R., G.D.G. and C.M.; methodology, M.M., M.R., G.D.G., F.C. and F.B.; investigation and data curation, M.M., M.R., G.D.G., F.C., F.B. and G.B.; writing—review and editing, M.M., G.D.G., C.M., F.C., F.B. and G.B. funding acquisition, M.M. and G.D.G. All authors have read and agreed to the published version of the manuscript.

**Funding:** This research was supported by the University of Urbino, Department research budget 2021, resp. M.M., and by University of Catania "Piano di incentivi per la ricerca d'Ateneo", 2020/2022, resp. G.d.G.

**Acknowledgments:** Sentinel-1 datasets were provided by the European Space Agency under a free, full and open data policy adopted for the Copernicus program. We thank four anonymous Reviewers for their fruitful suggestions that stimulated the significant improvement of the early manuscript.

**Conflicts of Interest:** The authors declare no conflict of interest.

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
