# Peer review of "Sentinel-1 Interferometry and UAV Aerial Survey for Mapping Coseismic Ruptures: Mts. Sibillini vs. Mt. Etna Volcano"

_remotesensing, doi:10.3390/rs15102514_

Round 1

Reviewer 1 Report

I have read this manuscript with interest and consider it worthy of publication after minor revisions. I have reported on the attached pdf some corrections to be made (mainly typos and spelling errors) and annotations (relating to minor specifications to be addressed).

Best regards.

Author Response

The manuscript has been changed according to the Reviewers notes

Reviewer 2 Report

The recognition and structural analysis of surface coseismic ruptures are essential to characterize seismgenic structures and improve regional seismic hazard assessments. This paper presents some methods (eg,. Sentinel-1 A/B Interferometric, UVA, InSAR, etc,.) to mapping coseismic ruptures, and selected Mts. Sibillini, Mt. Etna volcano as test objects, also give the detailed introduction, has important reference value. I also believe that the method used in this paper provides a convenient and high precision data acquisition method for the study of surface coseismic ruptures evolution and prediction.

However, the Discussion part (line 694) is rather lengthy and not easy for more readers to understand, so I suggestion further condensed and summarized into a few points.

Author Response

The manuscript has been modified according to the Reviewers notes

Reviewer 3 Report

Brief summary:

The manuscript presents results for detection of surface ruptures caused by earthquakes. Ruptures are first detected from satellite-based interferograms and then in more detail from UAV-based photogrammetric surveys. The results are compared with field measurements.

Broad comments:

1) It is stated that the aim of the manuscript is to "verify a new method" for recognition of ruptures. However, it is not clear what is new in the method. Has the method been proposed elsewhere if it is now only verified? It seems that an old method is verified or applied in two case studies.

2) The details of the methods are not fully explained but they may be found from the cited previous works of the authors and others. On the other hand, it would be difficult to reproduce the results, because the description in Section 2 is too general.

3) The manuscript is well structured. The figures are mainly good while some improvements are suggested below.

Detailed comments:

Lines 18, 126 and 807: Elastic deformation would return to the initial state when the stress releases. Do you mean frictionally plastic deformation instead of elastic deformation?

Lines 21 and 811: Which methods are compared? There seems to be no comparison between any methods in the manuscript.

Lines 14-30: Photogrammetry and UAV have not been mentioned in the abstract, although they appear in the title and keywords.

Lines 65 and 834: Geodynamic "contest" seems to be an incorrect word. Do you mean "context" or "content"?

Figure 2: PPRTK in phase 2 has not been explained.

Lines 139-140 and [38]: There is a contradiction: the text refers to interferograms but the analysis in [38] is based on seismic profiles.

Lines 218-219: AGL has not been explained.

Section 3.2 and Fig. 1b: The description of Mt. Etna includes several faults which are not named in Fig. 1b, such as Timbe fault system, Acireale fault, Linera-Santa Tecla fault, Santa Venerina fault, Catania and Terreforti anticlines, Aci Trezza fault. It would be easier to read the text if the faults were also shown in the figure.

Lines 352-357: It seems that the ruptures have been interpreted using the methods described in Section 2.2. Are the ruptures visible in the interferogram in Fig. 3 before the analysis?

Figure 3: Please include a reference where the tectonic features (normal faults, thrust faults) have been obtained from.

Figure 4a caption and line 267: There is a contradiction: the Amatrice 24 Aug 2016 earthquake was 6.2 Mw according to line 267 but 6 Mw according to Fig. 4a caption.

Figure 4: It is difficult to see any gradient lineaments in the areas where the yellow and red lines have been drawn in the interferometric slope map. Please show a close-up of the interferometric slope map without the yellow and red lines so that one can verify the presence of lineaments (edge lines) in the interferometric slope map.

Figure 5a: What is the blue line? It is missing from the legend.

Line 424 and Fig. 8: It is not clear how the rupture azimuth of 340 degrees can be determined from the displacements shown in Fig. 8. The rupture azimuths are shown in Fig. 5b.

Line 427: A slope is dimensionless, so it is not clear what a slope offset given in meters means. Is it the offset in the direction of the gradient? Why not give also a vertical offset?

Line 428 and Fig. 8: It is not clear how the sense of the slip (left lateral) can be derived from the (absolute) displacements shown in Fig. 8.

Lines 433-435: Where is the rupture aperture shown? One cannot verify the statement that the possible correlation between the aperture and displacement is not observable if the aperture is not shown.

Line 483: LGM has not been explained.

Line 505: Hd denotes horizontal displacement and not aperture.

Line 507: The distribution of fractures is visible in Fig. 9a and not in Fig. 9g, where only a piece of the road of less than 100 m in length is shown.

Figure 10: It seems that the black arrows indicate differences between the interferograms computed from the ascending and descending orbits. How are these differences interpreted in terms of the deformation field?

Captions of Figs. 13a and 14: There is a contradiction: The date (March 14, 2016) of the UAV survey was before the earthquake on December 26, 2018. One would expect that the survey was after the earthquake.

Line 796: Only spaceborne SAR images were used and no airborne ones.

Lines 816-819: The English language should be improved on these lines.

Author Response

(The authors gave the same response as above.)

Author Response

(The authors gave the same response as above.)

Round 2

Reviewer 4 Report

The Authors revised the paper taking into account my suggestions.